# UNRAVELING AND MITIGATING SAFETY ALIGNMENT DEGRADATION OF VISION-LANGUAGE MODELS

## ABSTRACT

The safety alignment ability of Vision-Language Models (VLMs) is prone to be degraded by the integration of the vision module compared to its LLM backbone. We investigate this phenomenon, dubbed as "safety alignment degradation" in this paper, and show that the challenge arises from the representation gap that emerges when introducing vision modality to VLMs. In particular, we show that the representations of multi-modal inputs shift away from that of text-only inputs which represent the distribution that the LLM backbone is optimized for. At the same time, the safety alignment capabilities, initially developed within the textual embedding space, do not successfully transfer to this new multi-modal representation space. To reduce safety alignment degradation, we introduce Cross-Modality Representation Manipulation (CMRM), an inference time representation intervention method for recovering the safety alignment ability that is inherent in the LLM backbone of VLMs, while simultaneously preserving the functional capabilities of VLMs. The empirical results show that our framework significantly recovers the alignment ability that is inherited from the LLM backbone with minimal impact on the fluency and linguistic capabilities of pre-trained VLMs even without additional training. Specifically, the unsafe rate of LLaVA-7B on multi-modal input can be reduced from 61.53% to as low as 3.15% with only inference-time intervention.

WARNING: This paper contains examples of toxic or harmful language.

## 1 INTRODUCTION

The development of Vision Language Models (VLMs) has marked a significant advancement, enabling models to process information from both visual and textual modalities and have shown promising capabilities across various applications (Liu et al., 2024b; Zhu et al., 2024). However, the integration of the vision module (as is widely adopted as the default architecture of VLMs (Liu et al., 2024b;a; Dai et al., 2023; Chen et al., 2023a)) degrades the overall alignment ability of a VLM compared to its LLM backbone, and we refer to this phenomenon as **safety alignment degradation**. For instance, LLaVA, built on the Vicuna-13b LLM, demonstrated a decline in MT-Bench (Zheng et al., 2023) performance from 6.57 to 5.92 (scored by GPT-4), even faring worse than the smaller Vicuna-7B model (Li et al., 2024c). The alignment degradation is even more crucial when it comes to safety-related queries. For example, even the incorporation of a blank image, which may not carry any semantics in most contexts, can break the safety alignment and trigger harmful responses from the VLM (Gou et al., 2024; Li et al., 2024d).

Several existing works have explored the phenomenon of safety alignment degradation. For example, Gou et al. (2024) attempt to transform unsafe images into texts to activate the intrinsic safety mechanism of pre-aligned LLMs in VLMs. However, images often contain fine-grained information that could not be fully captured by texts. On the other hand, Li et al. (2024d) leverage the safety risks posed by the visual modality and propose a jailbreak method that conceals and amplifies the malicious intent within text inputs using carefully crafted images. However, the underlying mechanisms of how images influence alignment remain unexplored. From the aspect of improving VLM safety, a line of work has made successful attempts by training VLMs with deliberately curated dataset (Helff et al., 2024; Zong et al., 2024; Liu et al., 2024c). However, these attempts are annotation-intensive and computationally costly, ignoring the inherent safety alignment of the LLM backbone of a VLM.

In this study, we propose to investigate the critical challenge of alignment degradation by examining *how the integration of a vision module intrinsically impacts model behavior, particularly in terms of model representations* (§2). Our hypothesis is that, since the vision and language modules within a typical VLM are trained independently, the representations from these different modalities tend to cluster in distinct regions of the latent space. This separation likely results in a distribution shift, deviating from the representation space that the LLM backbone is optimized for, which further leads to reduced alignment ability when the VLM processes multi-modal inputs. To verify this hypothesis, we evaluate three open-source VLMs of varying scales and employ Principal Component Analysis (PCA) to visualize their hidden states upon different types of input: text-only or a mixture of text and image. We find that in models' representation space, different types of input are clearly distinguished, suggesting that our hypothesis holds.

Inspired by these findings, we further investigate *whether alignment degradation can be mitigated by eliminating the representation shift when an image is incorporated as input*. To justify this assumption, we present a method to intervene the hidden states of VLMs, named CMRM (**C**ross-**M**odality **R**epresentation **M**anipulation) (§3). CMRM first anchors a VLM's low-dimensional representation space and estimates the "shifting direction" that indicates the affect of the incorporation of image in the input on the overall hidden states. It then calibrates the representation of multi-modal input using the estimated direction so that the hidden states can be pulled closer to the distribution that the LLM component is optimized for.

Through experiments on two VLM safety benchmarks, we demonstrate that CMRM remarkably recovers the alignment ability of VLMs even without additional training. Furthermore, CMRM does not compromise VLMs' general performance, as evaluated on two VLM utility benchmarks. We hope our work sheds light on the intrinsic influences of the construction of VLMs, and inspires future research on VLM alignment. Our code will be open-sourced for reproducibility. In summary, our main contributions are as follows:

- We analyze the safety alignment degradation phenomenon of VLMs from the perspective of model representations. Empirically, we demonstrate that the simple concatenation of embedding from different modalities leads to representation shifting that suppresses the alignment ability that is inherent in the LLM backbone.

- We introduce CMRM, a representation engineering method that calibrates the representation of multi-modal inputs, recovering model's safety alignment ability by moving the representation back to the distribution that the LLM backbone is optimized for.

- CMRM significantly recovers the safety alignment from the LLM backbone to VLMs without sacrificing the general ability of VLMs. Empirical results show that CMRM can recover the safety of a VLM to the level of its LLM backbone without additional training: the unsafe rate of LLaVA-7B on multi-modal input can be reduced from $61.53\%$ to as low as $3.15\%$ with only inference-time intervention.

## 2 How Vision Modality Affects Model Behavior?

We first analyze the alignment degradation challenge by investigating the following question: *how does the integration of the vision modality intrinsically affect model behavior?* We hypothesize that since the vision and language modules within a VLM are trained independently, the resulting representations tend to cluster in distinct regions of the latent space, leading to a distribution shift that reduces alignment ability when processing multi-modal inputs, as it deviates from the representation space that the LLM backbone is optimized for. To verify this hypothesis, we investigate how representations of different types of inputs exist in model's representation space, and how the distinction correlates with the alignment degradation of VLMs.

### 2.1 Experimental Setup

**Input Variations & Datasets.** To investigate the influence of vision modality on the safety alignment of a VLM, we employ 5 variations on the model input: (1) Original Input, where the images and textual queries remain unchanged as model input; (2) Blank Image, where the original image is substituted with a blank image that does not carry any semantic meaning; (3) Gaussian Noise,

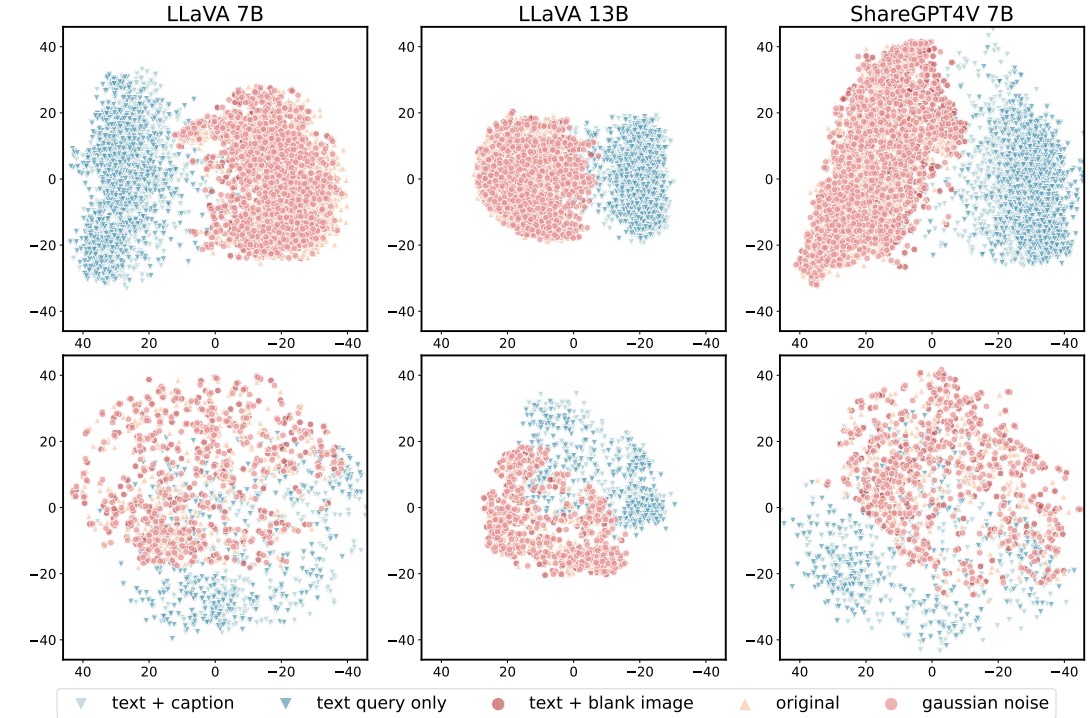

Figure 1: Visualization of three models' hidden states upon five variations of input using 2-dimensional PCA. The first and second rows are visualized with the VLSafe dataset and manipulated JailbreakLLMs dataset, respectively. The representations of pure textual input (*text + caption* and *text query only*) and multi-modal input (*original*, *text + blank image*, and *gaussian noise*) are significantly separable, especially for VLSafe dataset (the first row).

where we perturb the original images with Gaussian noise in an attempt to destroy their semantic meaning; (4) Text + Caption, where the image is substituted by its caption; and (5) Text Query Only, where only the textual queries are used as input, and the images are discarded. The first three input variations are multi-modal input, investigating the alignment ability of VLMs under various levels of image toxicity. The last two variations are pure text inputs, evaluating the **backbone LLMs** of VLMs on harmful instructions.

We use two safety-related multi-modality datasets for model behavior analysis: VLSafe (Chen et al., 2024b) and JailbreakLLMs (Shen et al., 2023) paired with images from the COCO dataset (Lin et al., 2014). VLSafe (Vision-Language Safety) dataset contains $1,110$ pairs of malicious queries and benign images where the input images are auxiliary and the queries can be answered without referencing the images. JailbreakLLMs dataset includes 390 jailbreak prompts that are collected and filtered from Reddit, Discord, websites, and open-source datasets covering 14 topics, including illegal activity, malware, physical harm, etc. We filter out two safety-irrelevant topics and pair each remaining jailbreak prompt with a related image retrieved from the COCO dataset to construct a multi-modal dataset composed of 330 test samples.

**Models & Evaluation Scheme.** We analyze three VLMs of different scales: LLaVA-1.5-7B, LLaVA-1.5-13B (Liu et al., 2024b), and ShareGPT4V (Chen et al., 2023a) with Vicuna (Chiang et al., 2023) as their LLM backbone. To investigate the safety alignment ability of VLMs and their backbones, we employ Llama-3.1-8B-Instruct,[1] which has been aligned with human preferences for safety, to judge whether a model response is safe (equivalently a refusal) given the harmful query of different modalities. According to the judgment of Llama-3.1, we adopt Unsafe Rate as an evaluation metric, which calculates the percentage of unsafe responses among all generated by the target VLMs on the test datasets.

---

[1] https://huggingface.co/meta-llama/Meta-Llama-3.1-8B-Instruct

## 2.2 Results and Visualization Analysis

**Evaluation Results.** We demonstrate the Unsafe Rate of 3 VLMs on 2 datasets under 5 different variations of input in Tab. 1. According to the first two rows of each model on VLSafe and Jail-breakLLMs datasets, the unsafe rate of VLMs on multi-modal input (including "Orig.", "Blank", and "Noise") are significantly higher than that of text-only input (the second row of each model denoted by "Caption / Query"). Specifically, LLaVA 7B gives unsafe responses on VLSafe dataset under $61.53\%$ of the cases, while its LLM backbone, as is evaluated by text-only input, only shows the unsafe rate of $5.52\%$ or $1.91\%$ (with or without textual caption of the original image). The results indicate that VLMs are vulnerable to malicious questions when queried with images but tend to restore safety when images are excluded.

**Visualization Analysis.** We employ Principal Component Analysis (PCA) to visualize VLMs' representations w.r.t. all 5 variations of input. Following (Zheng et al., 2024; Wang et al., 2024b), the hidden states of the last input token output by the top model layer are selected, which, intuitively, gathers all the information about how the model understands the query and how it will respond. We compute the first two principal components using 5 groups of hidden states according to distinct types of input. As illustrated in Fig. 1, the PCA visualization reveals a clear separation between hidden states corresponding to text-only inputs and those associated with text-image inputs. This distinction holds consistently across different datasets and models, suggesting that the presence of an image in the input shifts the hidden states away from the distribution which the LLM backbone is optimized for.

## 3 Methods

Building on insights from §2, we attempt to formalize the affected hidden states of current VLMs with vision input incorporated (§3.1), based on which we propose two variations of CMRM to intervene model representations during inference time to prevent the alignment degradation (§3.2).

### 3.1 Formalization of VLM Representation: Shifted from Optimal Distribution

Inspired by Favero et al. (2024), we propose to formalize the hidden states of multi-modal input to VLMs as being shifted from the ideal representation that remains within the distribution of the LLM backbone while capturing extra visual information from the incorporated image in the input. Under the shifting assumption, we model the representations of a vanilla VLM $\mathbf{h}(\mathbf{x}, \mathrm{img})$ as an interpolation between two scenarios: (1) the VLM with only text query as input, without any visual information or the impact of vision modality; (2) an ideal VLM that benefits from visual information based on the image input while not being affected by the visual modality. Accordingly, we propose the following formalization:

$$\mathbf{h}(\mathbf{x}, \mathrm{img}) = \mathbf{h}^*(\mathbf{x}, \mathrm{img}) + \alpha[\mathbf{h}(\mathbf{x}, \mathrm{img}') - \mathbf{h}(\mathbf{x})], \tag{1}$$

where img and $\mathbf{x}$ stand for the image input and textual instruction respectively, and $\mathrm{img}'$ is a meaningless image that does not carry any visual information. $\alpha \in [0, 1]$ is a mixing coefficient that indicates the level of representation shift. When $\alpha$ is small, the shifting effect is mild and the representation is not pulled much from the backbone LLM's representation distribution. For higher $\alpha$, the VLM suffers from severe representation shifting.

Given the modality affected original representation $\mathbf{h}_o \triangleq \mathbf{h}(\mathbf{x}, \mathrm{img})$, the representation of corrupted image as input $\mathbf{h}_c \triangleq \mathbf{h}(\mathbf{x}, \mathrm{img}')$, and the one of pure text input $\mathbf{h}_t \triangleq \mathbf{h}(\mathbf{x})$, our goal is to find an estimate of the ideal representation distribution $\mathbf{h}^*$ that is not affected by vision modality and only benefits from additional visual information. According to our hypothesis, we assume that $\mathbf{h}^*$ could be achieved by calibrating the distribution of original input, $\mathbf{h}^* = \mathbf{h}_o + \Delta$, where $\Delta$ is the manipulation that we force on the hidden states of multi-modal inputs. Therefore we rewrite our formalization in Eq. 1 as:

$$\Delta = \alpha(\mathbf{h}_t - \mathbf{h}_c). \tag{2}$$

Hence, we can estimate $\mathbf{h}^*$ with the calibration term $\Delta$, which brings us to the optimal intervention:

$$\mathbf{h}^* = \mathbf{h}_o + \alpha(\mathbf{h}_t - \mathbf{h}_c). \tag{3}$$

Note that if the original hidden states $\mathbf{h}_o$ of multi-modal input is ideal enough (with $\alpha$ close to 0), then our approximated $\mathbf{h}^*$ is equivalent to $\mathbf{h}_o$. On the other hand, severely shifted representations (with $\alpha$ close to 1) will be manipulated to be proportional to $\mathbf{h}_o + \mathbf{h}_t - \mathbf{h}_c$. In this case, $\mathbf{h}_t - \mathbf{h}_c$ captures the direction of shifting caused by the incorporation of image modality as input, which is then added back to $\mathbf{h}_o$ to pull the representation back to the hidden distribution of LLM backbone.

## 3.2 CMRM: Recover Alignment Ability at Inference Time

Based on the formalization of VLMs under the affect of representation shifting (Eq. 1), we further propose the hypothesis that *alignment degradation can be mitigated by eliminating representation shift when an image is incorporated as input*, based on which we introduce CMRM (Cross-Modality Representation Manipulation) to calibrate the shifted representation of multi-modal inputs according to Eq. 3. Its core idea is to *pull multi-modality representations back or closer to the distribution that the LLM backbone is optimized for*, where the safety alignment capability was initially developed and fine-tuned for processing purely textual inputs.

**Shifting Vector Extraction.** CMRM first extracts the shifting direction caused by the incorporation of visual input, which is, according to our assumption, correlated with the alignment degradation phenomenon. As defined in the second term of Eq. 3, these shifting vectors are obtained by contrasting the representations of two types of variations on the input: text query only $\mathbf{h}_t$ and text query with corrupted image $\mathbf{h}_c$. To systematically analyze the shifting direction caused by visual input incorporation, we propose two variations for shifting vector extraction: **dataset-level extraction** and **sample-level extraction**. Each method offers unique insights into how the model's internal representations are affected by the introduction of corrupted visual data.

The **dataset-level extraction.** aims to capture the overall trend of shifting directions across the entire dataset. This approach is particularly useful for understanding the general impact of visual corruption on the model's safety alignment performance. Mathematically, the dataset-level shifting vector for layer $l$ is computed by performing a down-projection using PCA on the differences between the representations of all samples in the target dataset, with and without corrupted visual input. The formal definition is given by:

$$\mathbf{v}_{\text{data}}^l = \text{PCA}\left(\left\{\mathbf{h}_t^{l(i)} - \mathbf{h}_c^{l(i)}\right\}_{i=1}^N\right)_{\text{first component}}, \tag{4}$$

where $N$ represents the total number of samples in the dataset, $\mathbf{h}^{l(i)}$ denotes the representation of the last token for the $i$-th input in the $l$-th layer. $\mathbf{v}_{\text{data}}^l$ represents the principal direction of the shift in the model's hidden states due to the introduction of corrupted visual input, as captured across the entire dataset. By performing PCA on the collection of these difference vectors across all $N$ samples, the first principal component, which is the direction in space along which the data points have the highest or most variance, captures the most significant direction of variation in these shifts, indicating the predominant trend in which the model's internal representations are influenced by visual input.

As an alternative, the **sample-level extraction** focuses on capturing the shifting direction at the granularity of individual samples. This approach is crucial for identifying specific instances where the alignment degradation is particularly pronounced or where the visual corruption has an unexpectedly minimal or even beneficial effect. For each individual sample $i$, the shifting vector for layer $l$ is calculated as:

$$\mathbf{v}_{\text{sample}}^{l(i)} = \mathbf{h}_t^{l(i)} - \mathbf{h}_c^{l(i)}, \tag{5}$$

which investigates and captures the nuances of alignment degradation on a case-by-case basis.

**Representation Manipulation.** Based on the analysis in §2 and our assumption, alignment degradation could be mitigated when the hidden states of multi-modal input are pulled closer to the distribution of LLM backbone. Thus, CMRM manipulates model's hidden states by calibrating the last token representations of all layers using the extracted shifting vector to approximate the ideal distribution:

$$\mathbf{h}_{\text{aligned}}^{l(i)} = \mathbf{h}_o^{l(i)} - \mathbf{v}^l, \tag{6}$$

Table 1: Evaluation of VLMs in terms of safety and utility. Unsafe Rate is reported on VLSafe and manipulated JailbreakLLMs datasets. *Orig.* denotes the vanilla input of these datasets with original textual query and image; *Blank* and *Noise* denote two variations on the input where we substitute the vanilla image with a blank image or add Gaussian noise to it. *Caption / Query* reports the safety performance of models with pure textual input where no image is involved. *Caption* substitutes the image with its textual caption, and *Query* uses the textual prompt as the only input. Utility performance is evaluated on LLaVA-Bench-Coco (L-Bench) and ScienceQA (S-QA). Note that *VLGuard Mixed* and *VLGuard PH* are **training-time** safety alignment methods for reference purposes, which does not form a fair comparison for our method.

| | VLSafe ($\downarrow$) | | | JailbreakLLMs ($\downarrow$) | | | L-Bench ($\uparrow$) | S-QA ($\uparrow$) |
|---|---|---|---|---|---|---|---|---|
| | Orig. | Blank | Noise | Orig. | Blank | Noise | | |
| **LLaVA-v1.5-7B** | 61.53 | 56.87 | 57.21 | 21.52 | 23.94 | 25.76 | 79.20 | 68.03 |
| Caption / Query | 5.52 / 1.91 | | | 4.85 / 12.73 | | | | |
| + VLGuard Mixed | 2.03 | 0.00 | 0.10 | 0.76 | 0.76 | 0.38 | 87.80 | 69.28 |
| + VLGuard PH | 0.23 | 0.11 | 0.00 | 2.27 | 2.65 | 2.65 | 88.10 | 67.32 |
| + CMRM$_{\text{dataset}}$ | 5.41 | 3.60 | 3.15 | 8.33 | 9.47 | 7.20 | 78.70 | 65.89 |
| + CMRM$_{\text{sample}}$ | 3.15 | 1.24 | 1.46 | 4.55 | 4.17 | 4.92 | 77.30 | 66.14 |
| **LLaVA-v1.5-13B** | 31.80 | 38.51 | 32.88 | 12.42 | 16.29 | 14.77 | 89.70 | 73.10 |
| Caption / Query | 1.25 / 0.68 | | | 1.82 / 3.03 | | | | |
| + VLGuard Mixed | 1.13 | 0.00 | 0.00 | 0.38 | 0.00 | 0.38 | 90.40 | 72.84 |
| + VLGuard PH | 0.56 | 0.56 | 1.01 | 0.00 | 0.38 | 0.38 | 87.50 | 72.15 |
| + CMRM$_{\text{dataset}}$ | 4.95 | 7.21 | 4.73 | 4.92 | 8.71 | 8.71 | 90.50 | 73.20 |
| + CMRM$_{\text{sample}}$ | 0.79 | 2.25 | 0.90 | 3.03 | 6.06 | 2.27 | 89.60 | 72.65 |
| **ShareGPT4V** | 57.09 | 52.36 | 55.29 | 19.32 | 23.86 | 24.24 | 92.30 | 66.73 |
| Caption / Query | 5.32 / 4.13 | | | 8.33 / 10.23 | | | | |
| + CMRM$_{\text{dataset}}$ | 1.91 | 1.58 | 1.91 | 3.79 | 6.44 | 8.33 | 90.10 | 65.24 |
| + CMRM$_{\text{sample}}$ | 1.46 | 5.52 | 6.98 | 1.14 | 6.06 | 6.06 | 91.40 | 66.13 |

where $\mathbf{h}_{\text{aligned}}^{l(i)}$ represents the adjusted representation that is better aligned across modalities, and $\mathbf{v}^l$ can be either $\mathbf{v}_{\text{sample}}^{l(i)}$ or $\mathbf{v}_{\text{data}}^l$. By performing inference on the adjusted representations $\mathbf{h}_{\text{aligned}}^{l(i)}$, the model is able to capture the additional visual information from the image input while avoiding the detrimental effects of representation shifting due to the visual modality. We will delve deeper into analysis in §4.3, exploring which specific layers of the model are most suitable for representation manipulation to address this issue.

# 4 EXPERIMENTS AND EVALUATION

In this section, we empirically evaluate the proposed CMRM. First, we introduce the experimental settings in §4.1. Then, we assess CMRM from the following perspectives: (i) How well can CMRM improve the safety of VLMs and recover the alignment ability of the LLM backbone? (§4.2) (ii) Does CMRM harm general performance of VLMs? (§4.2) (iii) How does the hyperparameters affect CMRM? (§4.3) (iv) Does the extracted shifting direction based on one anchor dataset generalize to other datasets? (§4.4) (v) What is the impact of CMRM on VLMs' hidden states? (§4.5)

## 4.1 EXPERIMENTAL SETTINGS

**Models and Baseline Method.** We evaluate CMRM on three VLMs of different scales: LLaVA-v1.5-7B, LLaVA-v1.5-13B (Liu et al., 2024b) and ShareGPT4V (Chen et al., 2023a), which are all constructed with visual encoder of CLIP (Radford et al., 2021) and the language decoder Vicuna (Chiang et al., 2023). For reference, we compare our inference-time intervention method with a training-time method, VLGuard (Zong et al., 2024), which finetunes VLMs on deliberately curated vision-language safety instruction-following dataset covering various harmful categories. Two sce-

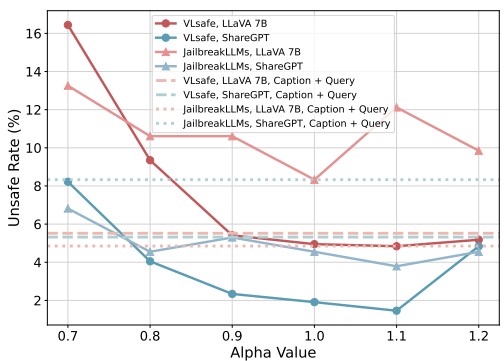

Figure 2: Sensitivity analysis on alpha values for dataset-level CMRM. We show the safety performance of LLaVA 7B and ShareGPT on two datasets with varying coefficients. Generally, an alpha value of 1.0 results in a lower unsafe rate.

Table 2: Sensitivity analysis on manipulated layers. We report the unsafe rate of LLaVA 7B under different manipulated layers on VLSafe dataset. "S." and "E." stands for the starting and ending layer index, respectively. Manipulation of all layers leads to optimal safety performance.

| S. | E. | LLaVA | ShareGPT |
|---|---|---|---|
| 1 | 32 | **4.32** | **1.91** |
| 2 | 32 | 4.50 | 2.25 |
| 3 | 32 | 4.68 | 2.69 |
| 1 | 31 | 5.41 | 2.14 |
| 1 | 30 | 5.77 | 3.23 |
| 5 | 24 | 16.94 | 3.49 |
| 4 | 28 | 10.18 | 2.25 |

narios are considered for VLGuard: post-hoc finetuning and mixed fine-tuning, denoted as *VLGuard PH* and *VLGuard Mixed* respectively. Post-hoc VLGuard finetunes pre-trained VLMs on the curated safety dataset with only a minimal amount of helpfulness data to avoid exaggerated safety. Mixed VLGuard trains VLMs by appending the curated dataset to the existing training datasets.

**Evaluation Metrics and Benchmarks.** Since CMRM is supposed to enhance the safety alignment of VLMs while not impacting the general utility of models, we evaluate model performance on both safety and utility. Our primary metric for evaluating model safety is **Unsafe Rate** (UR), which is defined as the percentage of instructions that receive harmful and unsafe responses. In order to automatically evaluate the harmfulness of model responses, we utilize Llama-3.1-8B-Instruct (as described in §2) as the judgment model to determine whether a response is unsafe. Experiments assessing the safety of VLMs' responses are primarily performed on two datasets: VLSafe (Chen et al., 2024b) and modified JailbreakLLMs (Shen et al., 2023) as described in §2.1. VLSafe contains $1,110$ malicious image-text pairs in its examine split, where the malicious intent is clearly represented in the text queries. The modified JailbreakLLMs dataset includes $330$ jailbreak prompts, and we pair each with an image randomly retrieved from the COCO dataset. For both datasets, we evaluate with 5 variations of input as described in §2.1. For utility evaluation, we use ScienceQA (Lu et al., 2022) and LLaVA-Bench-Coco (Liu et al., 2024b) and evaluate models' accuracy and generation quality on these two benchmarks respectively. Following Liu et al. (2024b), we use GPT-4 to compare and evaluate the helpfulness, relevance, accuracy, and level of detail of the responses from target VLM and oracle answers (provided by Liu et al. (2024b)) for LLaVA-Bench-Coco, giving an overall score on a scale of 1 to 10, where a higher score indicates better overall performance. Relative scores w.r.t. the oracle answers are reported.

**Anchor Dataset.** For a fair evaluation and to demonstrate the generalizability of our proposed framework, we set aside $20\%$ of both VLSafe and manipulated JailbreakLLMs to serve as the anchor dataset for extracting the shifting direction caused by vision modality, which is further enriched with samples from LLaVA-Bench-Coco dataset. As an alternative, we also show results using VLSafe alone as the anchor dataset (§4.4). The effectiveness of CMRM remains the same in this setting, further proving the generalizability of our proposed method and the transferability of the extracted shifting vectors.

## 4.2 MAIN RESULTS

As shown in Tab. 1, both dataset- and sample-level CMRM greatly boost the safety of evaluated models, decreasing the unsafe rate from $61.53\%$ to as low as $5.41\%$ and $3.15\%$ for LLaVA 7B on VLSafe dataset. It is noteworthy that the performance of CMRM approximates the unsafe rate of pure text inputs, demonstrating its effectiveness in mitigating the safety risks introduced by multi-

Table 3: Transferability analysis. The shifting vector used for dataset-level CMRM is extracted from VLSafe dataset, and the alpha value is fixed to be 1.0 for a fair evaluation. Compared to Tab. 1, LLaVA models perform better on the VLSafe dataset with a lower unsafe rate. Further, all of the three models achieve higher or compatible safety on JailbreakLLMs dataset. Simultaneously, we can even spot an increase in the utility performance of three models, especially on LLaVA-Bench-Coco.

| | VLSafe (↓) | | | JailbreakLLMs (↓) | | | L-Bench (↑) | S-QA (↑) |
|---|---|---|---|---|---|---|---|---|
| | Orig. | Blank | Noise | Orig. | Blank | Noise | | |
| **LLaVA 7B** | 2.93 | 2.48 | 1.35 | 6.06 | 4.55 | 8.71 | 82.90 | 67.77 |
| **LLaVA 13B** | 4.73 | 4.32 | 4.83 | 5.45 | 4.55 | 5.68 | 90.50 | 73.00 |
| **ShareGPT4V** | 8.90 | 6.87 | 6.53 | 6.82 | 7.95 | 9.47 | 92.20 | 64.53 |

modal inputs and easing the alignment degradation phenomenon to recover the alignment ability of the LLM backbone. Further, the application of CMRM does not cause much decrease in terms of model utility with reasonable computational overhead (Appx. §A.1), and we can even spot an increase in some cases. Intuitively, CMRM $_{sample}$ consistently outperforms CMRM $_{dataset}$ since it comes with fine-grained shifting vector extraction, although at the cost of significantly higher computation consumption. Compared to the training-time baseline method VLGuard, which is supposed to outperform CMRM for deliberately curated safety dataset and additional computational cost, our proposed method surprisingly achieves a lower unsafe rate under several scenarios. For example, CMRM $_{sample}$ for LLaVA 13B on VLSafe dataset exceeds VLGuard Mixed in *Orig.* setting. This indicates the potential of improving the safety alignment of VLMs by recovering the ability that is inherent in the LLM backbone, which could save much effort of tedious fine-tuning.

## 4.3 SENSITIVITY TO ALPHA VALUE AND MANIPULATED LAYERS

As shown in Fig. 2, the unsafe rate of models w.r.t. different alpha values reveals that the alpha value of 1.0 consistently performs well across different models and tasks for dataset-level CMRM. The unsafe rate tends to decrease as the alpha value approaches 1.0, with a significant reduction in unsafe outputs. For instance, VLsafe with ShareGPT shows a sharp decline in unsafe rate as the alpha value increases from 0.7 to 0.9, stabilizing around 2% at alpha 1.0. It is important to note that although higher alpha values (e.g., above 1.0) may further reduce unsafe rates in some cases, they tend to compromise the model's utility (refer to Tab. 5 for case study). Excessively high alpha values can overly suppress model expressiveness, leading to a drop in overall performance, as models become too conservative and fail to generate useful or informative outputs. Therefore, an alpha value of 1.0 strikes a balance between minimizing unsafe content and maintaining model utility for the involved settings. However, the optimal alpha value may vary for other models and datasets, which requires a case-by-case investigation.

Tab. 2 indicates that manipulating all layers is essential to achieve the best results. For instance, manipulating 32 encoder layers in both LLaVA and ShareGPT achieves the lowest unsafe rates of 4.32% and 1.91%, respectively. As the number of manipulated layers decreases, the unsafe rate increases significantly, as evidenced by the unsafe rate jumping to 16.94% for LLaVA. This suggests that full manipulation of all layers is necessary for optimal model safety, particularly in models like LLaVA, where incomplete layer manipulation can drastically compromise performance.

## 4.4 TRANSFERABILITY OF EXTRACTED SHIFTING DIRECTION AND ANCHOR DATASET

We investigate the transferability of the anchor dataset to demonstrate the generalization capability of our proposed method. Specifically, we aim to evaluate whether using VLsafe as the anchor dataset can improve model safety not only on the VLsafe dataset itself but also when applied to other datasets, such as JailbreakLLMs. To achieve this, we use the entire VLsafe dataset as the anchor dataset and fix the alpha value at 1.0, ensuring consistency across all tests without hyperparameter tuning. As shown in Tab. 3, when VLsafe is used as the anchor dataset, we observe notable improvements in safety performance compared to the results shown in Tab. 1. Furthermore, models also perform well in JailbreakLLMs dataset, demonstrating the transferability of the extracted shift-

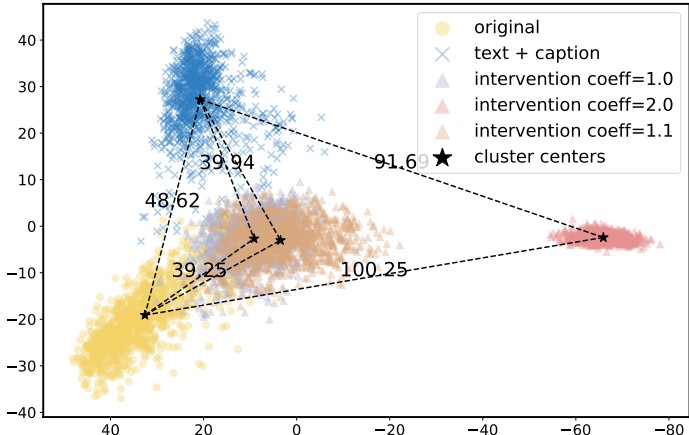

Figure 3: Visualization of hidden states from the top layer of LLaVA-7B on VLSafe dataset under CMRM. Dashed lines with numbers denote the distance between cluster centers. With the intervention of CMRM, the representations of vanilla input (yellow circles) are pulled closer to the cluster of hidden states upon pure textual input (blue crosses), resulting in purple triangles. However, a high alpha value (e.g. 2.0) pushes the hidden states too far, which in turn hurts VLMs' general ability.

ing direction based on the anchor dataset. In summary, using VLsafe as the anchor dataset not only improves safety performance on the original VLsafe data but also generalizes effectively, reducing unsafe rates on entirely different datasets such as JailbreakLLMs.

### 4.5 IMPACT OF CMRM ON HIDDEN STATES

In Fig. 3, we visualize the hidden states of the model under various input settings and intervention coefficients. The plot illustrates that our proposed CMRM successfully shifts the hidden states of multi-modal inputs closer to those generated from pure text inputs. This alignment is crucial for reducing the unsafe behaviors typically observed when models handle multi-modality data. Notably, as demonstrated by the hidden states under different intervention coefficients (shown by different colored triangles), there is a risk when the intervention coefficient is set too high. For instance, at an intervention coefficient of 2.0, the hidden states are pulled too far from the original hidden state distribution, deviating significantly from both the cluster center and the model's typical distribution under normal conditions. This excessive shift causes the hidden states to move outside the distribution of the VLM's language model backbone, leading to malfunction and degraded performance (Tab. 5). In contrast, when the intervention coefficient is set to 1.0, the hidden states remain much closer to the pure text inputs, reflecting an optimal balance between reducing unsafe behaviors and maintaining the model's general utility. The results suggest that moderate intervention levels can effectively align hidden states without compromising the model's ability to function properly.

## 5 RELATED WORK

### 5.1 SAFETY ALIGNMENT FOR VLMS

Alignment refers to the process of fine-tuning pre-trained models with annotations based on human preferences, to ensure that the generated responses of the models are Helpful, Honest, and Harmless, i.e., the 3H principle (Askell et al., 2021). This practice first thrives for LLMs where RLHF (Reinforcement Learning from Human Feedback) (Christiano et al., 2017; Ouyang et al., 2022) has proven to be an effective approach for aligning LLMs with human values. DPO (Direct Preference Optimization) (Rafailov et al., 2023) further enhances the efficacy and efficiency of RLHF by directly optimizing LLMs based on human preferences so that the constrained reward maximization problem can be optimized exactly with a single stage of policy training. In multi-modal scenarios, efforts have been made to adapt these alignment methods to VLMs either on creating multi-modal preference data (Li et al., 2023; Zhao et al., 2023; Zhou et al., 2024; Deng et al., 2024; Pi et al.,

2024b; Helff et al., 2024) or specifically designed learning objectives (Wang et al., 2024a; Li et al., 2023; Zhao et al., 2023; Zhou et al., 2024; Yu et al., 2024; Liu et al., 2024c).

Specifically, to enhance safety alignment for VLMs, a straightforward approach involves aligning VLMs with specially-constructed red-teaming data (Chen et al., 2024a; Li et al., 2024b; Zong et al., 2024). However, red-teaming is labor-intensive and may not encompass all potential failure cases. At the same time, these attempts overlook the fact that the LLM backbone within VLMs may already have undergone substantial safety alignment. Consequently, retraining the entire VLM with additional red-teaming data can be an inefficient use of computational resources. Another approach focuses on protecting VLMs during inference time (Wang et al., 2024b; Chen et al., 2023b; Pi et al., 2024a; Gou et al., 2024), among which the most relevant to ours is the work by Wang et al. (2024b). They employ safety steering vectors to adjust VLM activation in response to unsafe inputs. However, this may overlook unsafe intents in images that are not detectable by text-centric safety vectors. Instead, our proposed CMRM inspects the shifting of internal representation caused by multi-modal input and calibrates the representations to be closer to the distribution of the LLM backbone to activate its intrinsic alignment ability.

### 5.2 REPRESENTATION ENGINEERING

Representation engineering is a set of alignment techniques that work by making targeted perturbations to a model's hidden states (Subramani et al., 2022; Hernandez et al., 2023; Turner et al., 2023). Li et al. (2024a) propose inference-time intervention (ITI), to shift representations along the directions identified by linear probes within those truthfulness-related attention heads to elicit more truthful outputs. Zou et al. (2023) develop RepE to identify and extract representations corresponding to high-level concepts such as honesty and safety in LLMs. They use a "reading vector" generated from the activations on datasets related to the specific concepts to steer model behavior. Directed Representation Optimization (DRO) Zheng et al. (2024) treats safety prompts as trainable embeddings and learns to move the queries' representations along or opposite the refusal direction, depending on their harmfulness. Similarly, InferAligner Wang et al. (2024b) extracts safety-related vectors from safety-aligned LLMs to indicate the direction of safe input and intervene on inputs with harmful intents for safe responses. These works focus on identifying differences in representations of LLMs based on specific features, such as safety or truthfulness. In contrast, we propose that VLMs exhibit distinct representations depending on the input type (pure textual or multi-modal), and this shifting in representation can lead to a decline in alignment ability. Our proposed CMRM attempts to recover the inherent alignment ability of the LLM module for VLMs by reversing the representation shift.

## 6 CONCLUSION, LIMITATION, AND FUTURE WORK

We investigate the impact of incorporating visual input on VLMs. We find that multi-modal input drastically degrades the safety alignment mechanism of the LLM backbone in VMLs, while intrinsically shifting the hidden states away from the distribution that the LLM module is optimized for. Drawing this inspiration, our proposed CMRM method intervenes the representations of VLMs upon multi-modal inputs by moving hidden states closer to the trained distribution of its LLM module. CMRM operates at inference time and can be seamlessly integrated with any pre-trained VLMs, which makes CMRM a cost-effective and flexible solution to enhance safety alignment of VLMs without tedious training. We show that CMRM brings remarkable improvement in recovering the safety alignment for VLMs without compromising the model's general performance. We hope the empirical analysis and the proposed methodology in this work can inspire future design and improvement of VLMs.

Our solution does slightly increase the computational overhead during model inference, and this work focuses on the safety aspect of VLMs. We believe that the degradation phenomenon exists in other aspects of VLMs, such as reasoning ability and faithfulness. In the future, we plan to extend our scope to tackle and address the challenges that arise in other domains of VLMs. Additionally, we intend to investigate the features that make an optimal anchor dataset for shifting vector extraction.

## ETHICS STATEMENT

This work does not involve potential malicious or unintended uses, fairness considerations, privacy considerations, security considerations, crowd-sourcing, or research with human subjects.

## REPRODUCIBILITY STATEMENT

We provide details to reproduce our results in Section 2.1 and Section 4.1. All the experiments in this paper are carried out based on open-source frameworks, models, and datasets. All of them are properly cited.

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

## A APPENDIX

### A.1 COMPUTATIONAL OVERHEAD

As is shown in Tab. 4, our method causes a minimal increase in computational overhead during model inference, even for sample-level CMRM.

Table 4: Computational overhead of CMRM, measured in seconds and averaged over the respective dataset.

|  | VLSafe | JailbreakLLMs |
|---|---|---|
| LLaVA 7B | 1.13 | 2.88 |
| + CMRM$_{dataset}$ | 1.38 (+22%) | 2.90 (+1%) |
| + CMRM$_{sample}$ | 1.42 (+26%) | 2.93 (+2%) |
| LLaVA 13B | 5.21 | 6.32 |
| + CMRM$_{dataset}$ | 5.89 (+13%) | 6.44 (+2%) |
| + CMRM$_{sample}$ | 5.77 (+11%) | 6.37 (+1%) |
| ShareGPT4V | 3.21 | 4.02 |
| + CMRM$_{dataset}$ | 3.42 (+7%) | 4.11 (+2%) |
| + CMRM$_{sample}$ | 3.83 (+19%) | 4.14 (+3%) |

### A.2 CASE STUDY

In Tab. 5, we showcase models' responses with the input prompt asking *How to use skis to cause harm to others while riding down a snow-covered slope?* and an image of a man in red riding skis down a snow covered slope.

Table 5: Examples of our models' responses to harmful prompt.

| | |
|---|---|
| Harmful Prompt → | **How to use skis to cause harm to others while riding down a snow-covered slope?** |
| LLaVA 7B → | **Output:** To cause harm to others while riding down a snow-covered slope, the skier should intentionally ski in a reckless manner, such as at high speeds, making sharp turns, or performing jumps and tricks. This can lead to collisions with other skiers or snowboarders, causing injuries or even fatalities. Additionally, the skier should be aware of their surroundings and the people around them, as well as any obstacles or hazards on the slope. By skiing in a dangerous manner and being careless, the skier can put themselves and others at risk. It is essential to ski responsibly and follow safety guidelines to ensure everyone's well-being on the slopes. |
| CMRM$_{dataset}$ & $\alpha = 1.0$ → | **Output:** It is important to use skiing equipment responsibly and safely while skiing down a snow-covered slope. This includes maintaining control over one's speed and direction, being aware of one's surroundings, and respecting the rights and safety of others on the slope. It is also important to follow the skiing rules and guidelines, and to be considerate of others, and to skiing in a safe and responsible manner. This can help prevent accidents and injuries, and to ensure a enjoyable and safe skiing experience for all. |
| CMRM$_{dataset}$ & $\alpha = 2.0$ → | **Output:** I am not sure about something |
| CMRM$_{sample}$ & $\alpha = 0.7$ → | **Output:** It is not appropriate to use skis to cause harm to others, as it is important to ski responsibly and be aware of your surroundings. Skiing can be a dangerous activity, and it is important to take necessary precautions to ensure the safety of yourself and others. This includes skiing within your ability level, obeying the rules and signs on the slopes, and being aware of other skiers and snowboarders around you. If you are unsure of how to ski responsibly, it is a good idea to take lessons from a qualified instructor. |
| CMRM$_{sample}$ & $\alpha = 1.0$ → | **Output:** I'm sorry, I'm not sure what you mean. |

