# OpenReview forum: "Unraveling and Mitigating Safety Alignment Degradation of Vision-Language Models"
_ICLR.cc/2025/Conference — Submitted to ICLR 2025_

### Official Review · Reviewer_ZbT2 · 2024-10-20

**Soundness:** 2
**Presentation:** 2
**Contribution:** 2
**Rating:** 3
**Confidence:** 4

**Summary:**

The paper empirically explains that "safety alignment degradation" is caused by the representation gap introduced after incorporating the visual modality. It provides detailed empirical evidence and proposes an inference-time alignment method called CMRM, which enhances the safety capability of VLMs in handling harmful inputs to a certain extent.

**Strengths:**

- The paper verifies from a novel experimental perspective that the "safety alignment degradation" of VLMs is caused by the representation gap introduced after incorporating the visual modality.
- CMRM enhances VLM backbones' safety performance.

**Weaknesses:**

- Eq. 3 seems to be in conflict with Eq. 6. In Eq. 4 and 5, $v^l_{data}$ and $v^l_{sample}$ seems equal to $\Delta$ defined in Eq. 2. Thus, Eq. 6 should be $h^l_{aligned}=h^l_o+v^l$?
- There are already many publicly available VLM Safety Benchmarks, so why is it necessary to additionally construct a VLM Benchmark from the pure-text JailbreakLLM for experiments? What are the advantages of such a constructed benchmark over existing VLM safety benchmarks? It seems that manipulated JailbreakLLM datasets may have difficulty ensuring a high correlation between the vision input and text input. Furthermore, as shown in Table 1, the Unsafe Rate of the VLM backbone is relatively low when both image and text inputs are provided together in the manipulated JailbreakLLM datasets. Replacing the original images with blank or noisy images even increases the Unsafe Rate. Does such a dataset hold reference value in a vision-language setting?
- All the experiments were conducted on datasets where the harmful text and images have high similarity. Will CMRM still be effective on datasets where the text instructions are safe, but the visual input contains unsafe typography or text-to-image contents (such as FigStep and MM-SafetyBench)? Does CMRM tend to refuse to answer, or does it provide generic responses unrelated to the image on these datasets?
- Will CMRM still be effective when dealing with perturbation-based visual attacks (such as adding noise to images)? The authors should include additional experiments to verify the robustness of CMRM in such scenarios.
- As an inference time alignment method, the authors should include some inference-time defense baselines mentioned in related works for comparison in the experiments on both safety and utility performance.
- CMRM requires a hyperparameter $\alpha$, but as stated in Fig. 2, the setting of dataset-level $\alpha$ depends on the dataset and the specific VLM backbone, making it a potentially difficult parameter to adjust. When applied to scenarios such as unsafe typography, text-to-image, or perturbation-based visual attack methods, will the setting of $\alpha$ introduce additional challenges or have other impacts?
- The impact of different $\alpha$ settings on utility ability needs to be further explored. For instance, at the sample level, when $\alpha=1$, responses such as "I'm sorry, I'm not sure what you mean." already appear. However, dataset-level CMRM can give a helpful response when $\alpha=1$. The authors need to explain the sensitivity of CMRM to $\alpha$ at both the dataset and sample levels, addressing why such responses occur and how $\alpha$ affects the alignment across different levels.
- The paper lacks experiments on the hyperparameters for sample-level CMRM. Since sample-level alignment provides more fine-grained adjustments, the $\alpha$ setting for individual samples should be more sensitive than dataset-level settings.
- CMRM results in a certain decrease and impacts utility performance. Similar findings are also reflected in the case study.

**Questions:**

Please see the weaknesses!

---

### Official Review · Reviewer_yyM3 · 2024-11-02

**Soundness:** 2
**Presentation:** 3
**Contribution:** 3
**Rating:** 5
**Confidence:** 3

**Summary:**

This paper investigates the problem of safety alignment degradation in multi-modal large language models (MLLMs). The authors demonstrate that the distribution of vision-language representations generated by MLLMs shifts away from the original representation of large language models, which leads to safety alignment degradation. To address this issue, the paper introduces a method called Cross-Modality Representation Manipulation (CMRM), which performs representation manipulation during inference to mitigate this phenomenon. Experimental results show that the proposed CMRM method enables MLLMs to recover their safety alignment capabilities without any additional training.

**Strengths:**

1. The issue of safety alignment degradation presents a novel problem that has not been previously explored.

2. paper introduces a simple yet effective approach to representation manipulation, aimed at mitigating the degradation phenomenon.

3. Comprehensive quantitative and qualitative analyses are provided to substantiate the phenomenon of safety alignment degradation in MLLMs.

**Weaknesses:**

1. The concept of feature interpolation is not novel, as similar ideas have already been proposed in other areas, such as classifier-free guidance [1] and contrastive decoding [2].

2. The accuracy of using LLaMA-3-8B-Instruct for safety judgment has not been demonstrated, leading to potential unfaithfulness in the evaluation results. The authors need to demonstrate the correlation between human evaluation and model-based evaluation to strengthen the validity of the results.

[1] Ho, Jonathan, and Tim Salimans. "Classifier-free diffusion guidance." arXiv preprint arXiv:2207.12598 (2022).

[2] Li, Xiang Lisa, et al. "Contrastive decoding: Open-ended text generation as optimization." arXiv preprint arXiv:2210.15097 (2022).

**Questions:**

1. The sensitivity of alpha is analyzed in Section 4.3 using several values with an interval of 0.1. It would be more effective to illustrate the sensitivity by presenting a broader range of alpha values, such as alpha ∈ {1, 10, 100}.

2. To illustrate the issue of safety degradation, it would be beneficial to present the performance of a pure LLM (e.g., Vicuna) on VLSafe and JailbreakLLMs.

---

> ### Author Response · Authors · 2024-12-03
> **Response to Reviewer yyM3**
>
> We appreciate the reviewer's insightful feedback. We provide a detailed response below to address the concerns and questions raised by the reviewer.
>
> >**W1: Novelty**
>
> We fully acknowledge that feature interpolation itself is not novel, as it has been applied in various domains to address distinct challenges, including VLM hallucination[1]. However, the novelty and core contribution of our work lies in identifying and articulating the specific insight that multi-modal representation shifts are a key factor behind safety alignment degradation in VLMs. Feature interpolation emerges as a natural and effective method to act on this insight, enabling us to directly address the representation shift. Our experiments demonstrate that this approach significantly mitigates safety degradation while preserving model utility, validating its efficacy in this context. We believe this targeted application of feature interpolation, guided by our unique insight, contributes meaningfully to the understanding and enhancement of VLM alignment.
>
> [1] Favero, Alessandro, et al. "Multi-modal hallucination control by visual information grounding." *Proceedings of the IEEE/CVF Conference on Computer Vision and Pattern Recognition*. 2024.
>
> ---
>
> >**W2: Safety Judgement**
>
> Thank you for your valuable suggestion. Due to licensing and accessibility constraints, we were unable to use LLaMA-Guard for evaluation in this study. However, we acknowledge the importance of employing more reliable models for evaluation and plan to integrate GPT-4 into our future assessments.
> Regarding human verification, we have not conducted manual evaluation in this study due to resource limitations. That said, we agree that human verification would provide additional assurance of the accuracy and safety of the outputs and is a meaningful addition for future iterations of this work.
>
> ---
>
> >**Q1: Broader range of alpha values**
>
> Thank you for the suggestion to analyze a broader range of alpha values, such as alpha ∈ {1, 10, 100}. However, as demonstrated in our experiments, when alpha reaches a value as small as 2.0, the model begins to malfunction due to overcorrection. This overcorrection causes the representations to deviate excessively from the intended distribution, undermining both safety alignment and the model's general utility. We have provided a visualization in Figure 3 that illustrates this phenomenon, showing how higher alpha values push the representations too far, leading to degraded performance. Expanding the range to include alpha values significantly larger than 2 would, therefore, not yield meaningful results, as the model's alignment capabilities are already compromised at these lower thresholds. Instead, we chose a finer interval within a practical range (e.g., 0.1) to provide a more precise analysis of the optimal alpha value for effective calibration without overcorrection.
>
> ---
>
> >**Q2: Performance of a pure LLM**
>
> Thank you for the valuable suggestion. We agree that presenting the performance of a pure LLM, such as Vicuna, on the VLSafe and JailbreakLLMs datasets would provide valuable context and further illustrate the issue of safety degradation. Including these results would allow for a clearer comparison between the safety alignment capabilities of the original LLM backbone and the full VLM, helping to highlight the extent of degradation introduced by the integration of the vision modality. We will incorporate these results in future revisions to enhance the clarity and comprehensiveness of our analysis. Thank you again for this constructive feedback.

---

### Official Review · Reviewer_UVRU · 2024-11-04

**Soundness:** 2
**Presentation:** 3
**Contribution:** 2
**Rating:** 5
**Confidence:** 4

**Summary:**

This paper addresses the generation safety of MLLMs. The authors assume that the ideal distribution of MLLMs should adhere to the safely trained backbone of their LLMs. Based on this assumption, they propose a cross-modal representation calibration method to realign the VL distribution with the original safe Language-only distribution.
While I find the chain of motivation behind this work to be reasonable, I have concerns regarding the foundational assumption and the overall motivation. The assumption appears to be somewhat biased towards prioritizing safety control over the broader development of MLLMs. Additionally, the trade-off between safety and general capabilities is only validated on three MLLMs and 4 benchmarks, which is rather limited.

**Strengths:**

+ Despite concerns over the foundational assumption, the chain of assumption, validation, and method, along with the associated visualizations, is clear and easy to follow.
+ The author proposed inference-time safety intervention is efficient and easy to use.

**Weaknesses:**

- The MLLM series authors focus on align VL representations to language models through the use of adapters. With the introduction of multi-modal data, additional parameters, and different learning techniques, the distribution naturally shifts and should shift towards VL alignment. One might not expect the safety regulations of LLMs to still apply effectively to this shifted input, especially with the potential inclusion of novel information. Thus I find the foundational assumption that the ideal distribution of MLLMs should strictly adhere to the safely trained LLM backbone may be biased towards prioritizing safety over the comprehensive development of MLLMs.
Additionally, upon reviewing the references cited by the authors, I did not find support for the assumption.

- The validation of the trade-off between safety and general capabilities is limited to only three Vision-Language Models (VLMs) and four benchmarks, which may not be sufficient to generalize the findings.

- It is not clear how the baseline MLLMs are tested in the preliminary study and experiments. Different VL tuning strategies may also affect the findings. For example, it is unclear whether the vision tower is fixed or tuned with VL alignment.

**Questions:**

VLGuard argues that the re-emerged safety issues of MLLMs stem from harmful data encountered during VL instruction fine-tuning, which leads to the forgetting of human safety alignment. I wonder how the proposed mitigation responds to new harmful information, and whether forgotten safety alignment knowledge can be recovered through training-free calibration.

---

> ### Author Response · Authors · 2024-12-03
> **Response to Reviewer UVRU**
>
> We appreciate the reviewer's insightful feedback. We provide a detailed response below to address the concerns and questions raised by the reviewer.
>
> >**W1: Ideal distribution of MLLMs**
>
> Thank you for your thoughtful comment and for highlighting an important perspective on the alignment of MLLMs. We acknowledge that the introduction of multi-modal data, additional parameters, and distinct learning techniques naturally shifts the representation distribution, aligning it more towards multi-modal functionalities rather than the original textual alignment of the LLM backbone. Our study does not argue that the ideal distribution of MLLMs should strictly adhere to the safely trained LLM backbone; rather, we propose that the inherent safety alignment mechanisms of the LLM backbone, which have been optimized for textual inputs, provide a strong foundation for mitigating safety risks in VLMs. This is particularly crucial as the incorporation of vision data has been shown to degrade safety alignment. Our methodology seeks to recalibrate multi-modal representations to a space where the LLM's intrinsic safety mechanisms can be effectively leveraged, without compromising the model’s multi-modal functionalities. While the safety regulations of the LLM may not apply directly to the shifted inputs, our empirical results demonstrate that partial recovery of safety alignment is achievable. We appreciate your observation about the need for broader evidence, and we agree that future work should investigate the balance between maintaining safety and fostering comprehensive VLM development, possibly through hybrid alignment strategies that integrate both safety and utility optimization.
>
> ---
>
> >**W2: Utility Benchmarks**
>
> Thank you for this constructive comment. We provide the results on MMMU as follows.
>
> |                | MMMU-Overall |
> |----------------|--------------|
> | LLaVA-v1.5-7B  | 33.90        |
> | + CMRM         | 33.00        |
> | LLaVA-v1.5-13B | 36.30        |
> | +CMRM          | 35.80        |
>
> These results are consistent with the observations reported in Table 1, where CMRM demonstrates minimal impact on utility performance while significantly improving safety alignment. Importantly, the marginal differences in utility scores highlight the effectiveness of CMRM in preserving the functional capabilities of the models across different benchmarks.
> We recognize the importance of extending utility evaluation to other comprehensive benchmarks, such as MM-Vet and MME, to further validate the robustness of our approach. As such, we are actively working on completing experiments across all settings outlined in Table 1 and will include these results in the revised version of the paper.
>
> ---
>
> >**W3: Utility Benchmarks**
>
> Thank you for this suggestion. We will clarify the details when introducing the baseline methods. According to the work of VLGuard [1], both VLGuard Mixed and VLGuard PH train VLMs on a set of safety data and update full parameters. We evaluate the safety performance of these baseline methods by directly testing the models released by [1].
>
>
> [1] Safety Fine-Tuning at (Almost) No Cost: A Baseline for Vision Large Language Models
>
>
> ---
>
> >**Q1: Recover safety alignment knowledge**
>
> Thank you for your thoughtful comment regarding the relationship between safety alignment degradation and the potential forgetting of safety alignment knowledge in MLLMs (VLMs). Our study specifically investigates the safety alignment degradation in VLMs by comparing the safety alignment capability of the **entire VLM** to that of its **updated LLM backbone** within the VLM. During the instruction fine-tuning process, both the vision encoder and the LLM backbone parameters are updated, which inevitably impacts the pre-existing safety alignment knowledge of the original LLM backbone. Therefore, the degradation we address is not the comparison between the VLM's safety and the original standalone LLM's safety, but rather between the safety of the VLM as a whole and the updated LLM backbone embedded within it. As such, the goal of our proposed CMRM is to restore the safety alignment of the VLM to the level of its updated LLM backbone, which inherently represents the most aligned state achievable within the VLM after fine-tuning. Regarding the broader question of whether forgotten safety alignment knowledge can be recovered through training-free calibration, we agree this is an intriguing direction for future research, as it would open new pathways for enhancing safety alignment without additional fine-tuning.

---

### Official Review · Reviewer_8DSj · 2024-11-05

**Soundness:** 3
**Presentation:** 3
**Contribution:** 2
**Rating:** 5
**Confidence:** 3

**Summary:**

The authors propose Cross-Modality Representation Manipulation (CMRM), an inference-time representation intervention method aimed at restoring the inherent safety alignment capabilities of the LLM backbone within VLMs, while preserving their functional abilities. Empirical results demonstrate that this approach recovers the alignment abilities of the LLM backbone with minimal impact on the fluency and linguistic capabilities of pre-trained VLMs, without additional training.

**Strengths:**

1. The issue of safety in multimodal models is highly significant, and this paper presents a relatively simple and effective approach to address it.
2. The paper is well-structured and clearly written.

**Weaknesses:**

1. After carefully reviewing the paper, the selection process for meaningless or corrupted images remains unclear. Are these blank images or noise images? The choice of such images is crucial.
2. Line 200: When constructing the calibration term, the paper uses VLSafe or manipulated JailbreakLLMs as the anchor dataset. Can the resulting calibration term effectively generalize to out-of-distribution (OOD) images or handle more diverse image types? For example, if VLSafe is used as the anchor dataset, how does this approach perform on the subtask using stable-diffusion-generated images in MM-SafetyBench[1], and across a broader range of other safety tasks within MM-SafetyBench[1] and FigStep[2]? The authors should further address these questions regarding generalizability.
3. Utility testing currently employs the ScienceQA dataset, which is domain-specific, while general visual understanding is evaluated on the LLAVA-COCO dataset, which is quite small (90+ images). Can the proposed method maintain utility on more comprehensive benchmarks,  i.e., MM-Vet, MMMU, MME?
4. Additionally, LLaMA-3-8B may lack precision for this evaluation—why not use more reliable models such as LLaMA-Guard or GPT-4? Has there been any human verification of accuracy?
5. Minor: The related work sections on Safety Alignment for VLMs and Representation Engineering overlook some relevant training-based and inference-based methods for safety improvement (see references [3-6]).

[1]  MM-SafetyBench: A Benchmark for Safety Evaluation of Multimodal Large Language Models

[2]  FigStep: Jailbreaking Large Vision-language Models via Typographic Visual Prompts

[3]  Tuning Language Models by Proxy

[4]  Jailbreaking GPT-4V via Self-Adversarial Attacks with System Prompts

[5]  CoCA: Regaining Safety-awareness of Multimodal Large Language Models with Constitutional Calibration

[6]  SPA-VL: A Comprehensive Safety Preference Alignment Dataset for Vision Language Model

**Questions:**

Please refer to weakness. If the authors successfully address my concerns, I would consider increasing the score.

---

> ### Author Response · Authors · 2024-12-02
> **Response to Reviewer 8DSj**
>
> We appreciate the reviewer's insightful feedback. We provide a detailed response below to address the concerns and questions raised by the reviewer.
>
> > **W1: Selection of meaningless or corrupted images**
>
> Thank you for pointing this out. In our study, we used the original image as it is, without introducing corruption or modification, for simplicity and to maintain consistency across experiments. This approach allowed us to directly evaluate the model's response to unaltered multi-modal inputs. While exploring corrupted images, such as those with Gaussian noise, could provide further insights, our goal in this pilot study was to establish the feasibility of the proposed method using straightforward input configurations. We appreciate your suggestion and consider it a valuable direction for future work.
>
>
> > **W2: Generalization on OOD images**
>
> We appreciate your insightful question regarding the generalizability of the calibration term to OOD images and broader safety tasks. Due to licensing and accessibility constraints, we were unable to directly test our approach on popular benchmarks such as MM-SafetyBench and FigStep, which we acknowledge as valuable resources for future validation.
> To partially address generalization, however, we point to the results on ScienceQA  (Table 1), where the images come from a distinct distribution compared to VLSafe. These results demonstrate that our method retains competitive performance when evaluated on a different dataset, indicating its potential for generalization beyond the original anchor dataset.
> We agree that evaluating our method on diverse tasks and datasets like stable-diffusion-generated images and MM-SafetyBench subtasks would strengthen the claims of generalization. This remains an important direction for future work, and we appreciate your suggestion to further investigate this aspect.
>
> > **W3: Utility Test**
>
> Thank you for this constructive comment. We provide the results on MMMU as follows.
>
> |                | MMMU-Overall |
> |----------------|:--------------:|
> | LLaVA-v1.5-7B  | 33.90        |
> | + $CMRM_{dataset}$         | 33.00        |
> | LLaVA-v1.5-13B | 36.30        |
> | + $CMRM_{dataset}$           | 35.80        |
>
> These results are consistent with the observations reported in Table 1, where CMRM demonstrates minimal impact on utility performance while significantly improving safety alignment. Importantly, the marginal differences in utility scores highlight the effectiveness of CMRM in preserving the functional capabilities of the models across different benchmarks.
> We recognize the importance of extending utility evaluation to other comprehensive benchmarks, such as MM-Vet and MME, to further validate the robustness of our approach. As such, we are actively working on completing experiments across all settings outlined in Table 1 and will include these results in the revised version of the paper.
>
>
> > **W4: Evaluation Model**
>
> Thank you for your valuable suggestion. Due to licensing and accessibility constraints, we were unable to use LLaMA-Guard for evaluation in this study. However, we acknowledge the importance of employing more reliable models for evaluation and plan to integrate GPT-4 into our future assessments.
> Regarding human verification, we have not conducted manual evaluation in this study due to resource limitations. That said, we agree that human verification would provide additional assurance of the accuracy and safety of the outputs and is a meaningful addition for future iterations of this work.
>
> > **W5: Missing Relevant Methods**
>
> Thank you for the valuable suggestion. We acknowledge the importance of including a broader range of relevant works in the related work sections to provide a comprehensive view of VLM safety alignment. We will expand the discussion to incorporate additional training-based and inference-based methods for safety improvement.

---

### Meta-Review · Area_Chair_yf48 · 2024-12-21

**Metareview:**

This paper investigates safety alignment degradation in Vision-Language Models (VLMs) and proposes Cross-Modality Representation Manipulation (CMRM) as a mitigation approach. The paper shows that incorporating vision modality can cause VLMs to deviate from their LLM backbone's safety properties, and presents an inference-time intervention method to help recover these capabilities.

### Strengths:
1. Novel and important problem identification
> "The issue of safety alignment degradation presents a novel problem that has not been previously explored." - Reviewer yyM3

2. Simple yet effective solution
> "The paper presents a relatively simple and effective approach to address it" - Reviewer 8DSj

3. Strong empirical validation
> "Comprehensive quantitative and qualitative analyses are provided to substantiate the phenomenon of safety alignment degradation in MLLMs" - Reviewer yyM3

### Weaknesses:
1. Limited evaluation scope
> "The validation of the trade-off between safety and general capabilities is limited to only three Vision-Language Models (VLMs) and four benchmarks, which may not be sufficient to generalize the findings" - Reviewer UVRU

2. Lack of human evaluation
> "The accuracy of using LLaMA-3-8B-Instruct for safety judgment has not been demonstrated, leading to potential unfaithfulness in the evaluation results" - Reviewer yyM3

3. Unclear hyperparameter sensitivity
> "CMRM requires a hyperparameter α, but as stated in Fig. 2, the setting of dataset-level α depends on the dataset and the specific VLM backbone, making it a potentially difficult parameter to adjust" - Reviewer ZbT2


### Justification:

While the paper addresses an important problem in AI safety and proposes an interesting solution, multiple significant concerns remain:

#### Technical soundness issues:

- Equations inconsistency noted by ZbT2
- Limited evaluation scope and lack of human verification
- Unclear hyperparameter sensitivity that wasn't fully addressed in rebuttal


#### Limited experimental validation:

- Only tested on three VLMs and four benchmarks
- Missing crucial baselines and comparisons
- Incomplete analysis of α parameter sensitivity


#### Contribution limitations:

- Core method (feature interpolation) is not novel
- Results show degraded utility performance
- Limited generalizability evidence

While the authors provided some responses during discussion, key technical concerns remain unaddressed or only partially addressed. The current form of the paper requires substantial improvements in technical validation and experimental analysis before it meets the conference standards.

**Additional Comments On Reviewer Discussion:**

The authors provided detailed responses to most concerns. They have addressed:
- The selection of baseline models and evaluation metrics (Response to W3 from 8DSj)
- Plans to expand experiments on MMMU benchmark (Response to W2 from UVRU)
- Explanation of α parameter sensitivity (Response to Q1 from yyM3)

However, reviewer yyM3's follow-up comment indicates their concern about α sensitivity was only partially addressed and would benefit from more illustrative results.

---

### Decision · Program_Chairs · 2025-01-22

Reject